# Affine-Invariant Online Optimization
# and the Low-rank Experts Problem

**Tomer Koren**
Google Brain
1600 Amphitheatre Pkwy
Mountain View, CA 94043
tkoren@google.com

**Roi Livni**
Princeton University
35 Olden St.
Princeton, NJ 08540
rlivni@cs.princeton.edu

## Abstract

We present a new affine-invariant optimization algorithm called *Online Lazy Newton*. The regret of Online Lazy Newton is independent of conditioning: the algorithm's performance depends on the best possible preconditioning of the problem in retrospect and on its *intrinsic* dimensionality. As an application, we show how Online Lazy Newton can be used to achieve an optimal regret of order $\sqrt{rT}$ for the low-rank experts problem, improving by a $\sqrt{r}$ factor over the previously best known bound and resolving an open problem posed by Hazan et al. [15].

## 1 Introduction

In the online convex optimization setting, a learner is faced with a stream of $T$ convex functions over a bounded convex domain $\mathcal{X} \subseteq \mathbb{R}^d$. At each round $t$ the learner gets to observe a single convex function $f_t$ and has to choose a point $\mathbf{x}_t \in \mathcal{X}$. The aim of the learner is to minimize the cumulative $T$-round regret, defined as

$$\sum_{t=1}^{T} f_t(\mathbf{x}_t) - \min_{\mathbf{x} \in \mathcal{X}} \sum_{t=1}^{T} f_t(\mathbf{x}).$$

In this very general setting, Online Gradient Descent [25] achieves an optimal regret rate of $\Theta(GD\sqrt{T})$, where $G$ is a bound on the Lipschitz constants of the $f_t$ and $D$ is a bound on the diameter of the domain, both with respect to the Euclidean norm. For simplicity, let us restrict the exposition to linear losses $f_t(\mathbf{x}_t) = \mathbf{g}_t^\mathsf{T} \mathbf{x}_t$, in which case $G$ bounds the maximal Euclidean norm $\|\mathbf{g}_t\|$; it is well known that the general convex case can be easily reduced to this case.

One often useful way to obtain faster convergence in optimization is to employ preconditioning, namely to apply a linear transformation $P$ to the gradients before using them to make update steps. In an online optimization setting, if we have had access to the best preconditioner in hindsight we could have achieved a regret rate of the form $\Theta(\inf_P G_P D_P \sqrt{T})$, where $D_P$ is the diameter of the set $P \cdot \mathcal{X}$ and $G_P$ is a bound on the norm of the conditioned gradients $\|P^{-1}\mathbf{g}_t\|$. We shall thus refer to the quantity $G_P D_P$ as the *conditioning* of the problem when a preconditioner $P$ is used.

In many cases, however, it is more natural to directly assume a bound $|\mathbf{g}_t^\mathsf{T} \mathbf{x}_t| \leq C$ on the magnitude of the losses rather than assuming the bounds $D$ and $G$. In this case, the condition number need not be bounded and typical guarantees of gradient-based methods such as online gradient descent do not directly apply. In principle, it is possible to find a preconditioner $P$ such that $G_P D_P = O(C\sqrt{d})$, and if one further assumes that the intrinsic dimensionality of the problem (i.e., the rank of the loss vectors $\mathbf{g}_1, \ldots, \mathbf{g}_T$) is $r \ll d$, the conditioning of the optimization problem can be improved to $O(C\sqrt{r})$. However, this approach requires one to have access to the transformation $P$, which is typically data-dependent and known only in retrospect.

In this paper we address the following natural question: can one achieve a regret rate of $O(C\sqrt{rT})$ without the explicit prior knowledge of a good preconditioner $P$? We answer to this question in the affirmative and present a new algorithm that achieves this rate, called Online Lazy Newton. Our algorithm is a variant of the Online Newton Step algorithm due to Hazan et al. [14], that employs a lazy projection step. While the Online Newton Step algorithm was designed to exploit curvature in the loss functions (specifically, a property called exp-concavity), our adaptation is aimed at general—possibly even linear—online convex optimization and exploits latent low-dimensional structure. It turns out that this adaptation of the algorithm is able to achieve $O(C\sqrt{rT})$ regret up to a small logarithmic factor, without any prior knowledge of the optimal preconditioner. A crucial property of our algorithm is its affine-invariance: Online Lazy Newton is invariant to any affine transformation of the gradients $\mathbf{g}_t$, in the sense that running the algorithm on gradients $\mathbf{g}'_t = P^{-1}\mathbf{g}_t$ and applying the inverse transformation $P$ on the produced decisions results with the same decisions that would have been obtained by applying the algorithm directly to the original vectors $\mathbf{g}_t$.

As our main application, we establish a new regret rate for the *low rank experts* problem, introduced by Hazan et al. [15]. The low rank experts setting is a variant of the classical prediction with expert advice problem, where one further assumes that the experts are linearly dependent and their losses span a low dimensional space of rank $r$. The challenge in this setting is to achieve a regret rate which is independent of number of experts, and only depends on their rank $r$. In this setting, Hazan et al. [15] proved a lower bound of $\Omega(\sqrt{rT})$ on the regret, but fell short of providing a matching upper bound and only gave an algorithm achieving a suboptimal $O(r\sqrt{T})$ regret bound. Applying the Online Lazy Newton algorithm to this problem, we are able to improve upon the latter bound and achieve a $O(\sqrt{rT\log T})$ regret bound, which is optimal up to a $\sqrt{\log T}$ factor and improves upon the prior bound unless $T$ is exponential in the rank $r$.

## 1.1 Related work

Adaptive regularization is an important topic in online optimization that has received considerable attention in recent years. The AdaGrad algorithm presented in [9] (as well as the closely related algorithm was analyzed in [22]) dynamically adapts to the geometry of the data. In a sense, AdaGrad learns the best preconditioner from a trace-bounded family of Mahalanobis norms. (See Section 2.2 for a detailed discussion and comparison of guarantees). The MegaGrad algorithm of van Erven and Koolen [23] uses a similar dynamic regularization technique in order to obliviously adapt to possible curvature in the loss functions. Lower bounds for preconditioning when the domain is unbounded has been presented in [7]. These lower bounds are inapplicable, however, once losses are bounded (as assumed in this paper). More generally, going beyond worst case analysis and exploiting latent structure in the data is a very active line of research within online learning. Work in this direction include adaptation to stochastic i.i.d data (e.g., [11, 12, 20, 8]), as well as the exploration of various structural assumptions that can be leveraged for better guarantees [4, 12, 13, 5, 19].

Our Online Lazy Newton algorithm is a part of a wide family of algorithms named Follow the Regularized Leader (FTRL). FTRL methods choose at each iteration the minimizer of past observed losses with an additional regularization term [16, 12, 21]. Our algorithm is closely related to the *Follow The Approximate Leader* (FTAL) algorithm presented in [14]. The FTAL algorithm is designed to achieve logarithmic regret rate for exp-concave problems, exploiting the curvature information of such functions. In contrast, our algorithm is aimed for optimizing general convex functions with possibly no curvature; while FTAL performs FTL over the second-order approximation of the functions, Online Lazy Newton instead utilizes a first-order approximation with an additional rank-one quadratic regularizer. Another algorithm closely related to ours is the Second-order Perceptron algorithm of Cesa-Bianchi et al. [3] (which in turn is closely related to the Vovk-Azoury-Warmuth forecaster [24, 1]), which is a variant of the classical Perceptron algorithm adapted to the case where the data is "skewed", or ill-conditioned in the sense used above. Similarly to our algorithm, the Second-order Perceptron employs adaptive whitening of the data to address its skewness. Finally, the SON algorithm, proposed in [18] is an enhanced version of Online Newton Step which utilizes sketching to improve over previous second order online learning algorithms. Similar to our paper, they propose a version that is completely invariant to linear transformations. Their regret bound (for our setting) depends linearly on the dimension of the ambient space, and quadratic in the rank of the loss matrix. In contrast, our regret bounds do not depend on the dimension of the ambient space and are linear in the rank of the loss matrix – two properties that are necessary in order to achieve optimal regret bound for the low rank expert problem.

This work is highly inspired and motivated by the problem of *low rank experts* to which we give an optimal algorithm. The problem was first introduced in [15] where the authors established a regret rate of $\widetilde{O}(r\sqrt{T})$, where $r$ is the rank of the experts' losses, which was the first regret bound in this setting that did not depend on the total number of experts. The problem has been further studied and investigated by Cohen and Mannor [6], Barman et al. [2] and Foster et al. [10]. Here we establish the first tight upper bound (up to logarithmic factor) that is independent of the total number of experts $N$.

## 2  Setup and Main Results

We begin by recalling the standard framework of Online Convex Optimization. At each round $t = 1, \ldots, T$ a learner chooses a decision $\mathbf{x}_t$ from a bounded convex subset $\mathcal{X} \subseteq \mathbb{R}^d$ in $d$-dimensional space. An adversary then chooses a convex cost function $f_t$, and the learner suffers a loss of $f_t(\mathbf{x}_t)$. We measure the performance of the learner in terms of the regret, which is defined as the difference between accumulated loss incurred by the learner and the loss of the best decision in hindsight. Namely, the $T$-round regret of the learner is given by

$$\text{Regret}_T = \sum_{t=1}^{T} f_t(\mathbf{x}_t) - \min_{\mathbf{x} \in \mathcal{X}} \sum_{t=1}^{T} f_t(\mathbf{x}).$$

One typically assumes that the diameter of the set $\mathcal{X}$ is bounded, and that the convex functions $f_1, \ldots, f_T$ are all Lipschitz continuous, both with respect to certain norms on $\mathbb{R}^d$ (typically, the norms are taken as dual to each other). However, a main point of this paper is to refrain from making explicit assumptions on the geometry of the optimization problem, and design algorithms that are, in a sense, oblivious to it.

**Notation.**  Given a positive definite matrix $A \succ 0$ we will denote by $\|\cdot\|_A$ the norm induced by $A$, namely, $\|\mathbf{x}\|_A = \sqrt{\mathbf{x}^\top A \mathbf{x}}$. The dual norm to $\|\cdot\|_A$ is defined by $\|\mathbf{g}\|_A^* = \sup_{\|\mathbf{x}\|_A \leq 1} |\mathbf{x}^\top \mathbf{g}|$ and can be shown to be equal to $\|\mathbf{g}\|_A^* = \|\mathbf{g}\|_{A^{-1}}$. Finally, for a non–invertible matrix $A$, we denote by $A^\dagger$ its Moore–Penrose psuedo inverse.

### 2.1  Main Results

Our main results are affine invariant regret bounds for the Online Lazy Newton algorithm, which we present below in Section 3. We begin with a bound for linear losses that controls the regret in terms of the intrinsic dimensionality of the problem and a bound on the losses.

**Theorem 1.** *Consider the online convex optimization setting with linear losses $f_t(\mathbf{x}) = \mathbf{g}_t^\top \mathbf{x}$, and assume that $|g_t^\top \mathbf{x}| \leq C$ for all $t$ and $\mathbf{x} \in \mathcal{X}$. If Algorithm 1 is run with $\eta < 1/C$, then for every $H \succ 0$ the regret is bounded as*

$$\text{Regret}_T \leq \frac{4r}{\eta} \log\left(1 + \frac{(D_H G_H T)^2}{r}\right) + 3\eta\left(1 + \sum_{t=1}^{T} |g_t^\top \mathbf{x}^\star|^2\right), \tag{1}$$

*where $r = \text{rank}(\sum_{t=1}^{T} \mathbf{g}_t \mathbf{g}_t^\top) \leq d$ and*

$$D_H = \max_{\mathbf{x}, \mathbf{y} \in \mathcal{X}} \|\mathbf{x} - \mathbf{y}\|_H, \qquad G_H = \max_{1 \leq t \leq T} \|\mathbf{g}_t\|_H^*.$$

By a standard reduction, the analogue statement for convex losses holds, as long as we assume that the dot-products between gradients and decision vectors are bounded.

**Corollary 2.** *Let $f_1, \ldots, f_T$ be an arbitrary sequence of convex functions over $\mathcal{X}$. Suppose Algorithm 1 is run with $1/\eta > \max_t \max_{\mathbf{x} \in \mathcal{X}} |\nabla_t^\top \mathbf{x}_t|$. Then, for every $H \succ 0$ the regret is bounded as*

$$\text{Regret}_T \leq \frac{4r}{\eta} \log\left(1 + \frac{(D_H G_H T)^2}{r}\right) + 3\eta\left(1 + \sum_{t=1}^{T} |\nabla_t^\top \mathbf{x}^\star|^2\right), \tag{2}$$

*where $r = \text{rank}(\sum_{t=1}^{T} \nabla_t \nabla_t^\top) \leq d$ and*

$$D_H = \max_{\mathbf{x}, \mathbf{y} \in \mathcal{X}} \|\mathbf{x} - \mathbf{y}\|_H, \qquad G_H = \max_{1 \leq t \leq T} \|\nabla_t\|_H^*.$$

In particular, we can use the theorem to show that as long as we bound $|\nabla f(\mathbf{x}_t)^\mathsf{T}\mathbf{x}_t|$ by a constant—a significantly weaker requirement than assuming bounds on the diameter of $\mathfrak{X}$ and on the norms of the gradients—one can find a norm $\|\cdot\|_H$ for which the quantities $D_H$ and $G_H$ are properly bounded. We stress again that, importantly, Algorithm 1 need not know the matrix $H$ in order to achieve the corresponding bound.

**Theorem 3.** *Assume that*

$$\max_{1 \le t \le T} \max_{\mathbf{x} \in \mathfrak{X}} |\nabla_t^\mathsf{T}\mathbf{x}| \le C.$$

*Let* $r = \mathrm{rank}(\sum_{t=1}^T \nabla_t \nabla_t^\mathsf{T}) \le d$, *and run Algorithm 1 with* $\eta = \Theta\big(\sqrt{r\log(T)/T}\big)$. *The regret of the algorithm is then at most* $O\big(C\sqrt{rT\log T}\big)$.

## 2.2 Discussion

It is worth comparing our result to previously studied adaptive regularization algorithms techniques. Perhaps the most popular gradient method that employs adaptive regularization is the AdaGrad algorithm introduced in [9]. The AdaGrad algorithm enjoys the regret bound depicted in Eq. (3). It is competitive with any fixed regularization matrix $S > 0$ such that $\mathrm{Tr}(S) \le d$:

$$\mathrm{Regret}_T(\text{AdaGrad}) \;=\; O\!\left(\sqrt{d} \inf_{\substack{S > 0, \\ \mathrm{Tr}(S) \le d}} \sqrt{\sum_{t=1}^T \|\mathbf{x}^\star\|_2^2 \, \|\nabla_t\|_{S^*}^2}\right), \tag{3}$$

$$\mathrm{Regret}_T(\text{OLN}) \;\;\;=\; \widetilde{O}\!\left(\sqrt{r} \inf_{S > 0} \sqrt{\sum_{t=1}^T \|\mathbf{x}^\star\|_S^2 \, \|\nabla_t\|_{S^*}^2}\right). \tag{4}$$

On the other hand, for every matrix $S > 0$ by the generalized Cauchy-Schwartz inequality we have $\|\nabla_t^\mathsf{T}\mathbf{x}^\star\| \le \|\nabla_t\|_S^* \|\mathbf{x}^\star\|_S$. Plugging this into Eq. (2) and a proper tuning of $\eta$ gives a bound which is competitive with *any* fixed regularization matrix $S > 0$, depicted in Eq. (4).

Our bound improves on AdaGrad's regret bound in two ways. First, the bound in Eq. (4) scales with the *intrinsic dimension* of the problem: when the true underlying dimensionality of the problem is smaller than the dimension of the ambient space, Online Lazy Newton enjoys a superior regret bound. Furthermore, as demonstrated in [15], the dependence of AdaGrad's regret on the ambient dimension is not an artifact of the analysis, and there are cases where the actual regret grows polynomially with $d$ rather than with the true rank $r \ll d$.

The second case where the Online Lazy Newton bound can be superior over AdaGrad's is when there exists a conditioning matrix that improves not only the norm of the gradients with respect to the Euclidean norm, but also that the norm of $\mathbf{x}^\star$ is smaller with respect to the optimal norm induced by $S$. More generally, whenever $\sum_{t=1}^T (\nabla_t^\mathsf{T}\mathbf{x}^\star)^2 < \sum_{t=1}^T \|\nabla_t\|_S^2 \|\mathbf{x}^\star\|_2^2$, and in particular when $\|\mathbf{x}^\star\|_S < \|\mathbf{x}^\star\|_2$, Eq. (4) will produce a tighter bound than the one in Eq. (3).

## 3 The Online Lazy Newton Algorithm

We next present the main focus of this paper: the affine-invariant algorithm Online Lazy Newton (OLN), given in Algorithm 1. The algorithm maintains two vectors, $\mathbf{x}_t$ and $\mathbf{y}_t$. The vector $\mathbf{y}_t$ is updated at each iteration using the gradient of $f_t$ at $\mathbf{x}_t$, via $\mathbf{y}_t = \mathbf{y}_{t-1} - \nabla_t$ where $\nabla_t = \nabla f_t(\mathbf{x}_t)$. The vector $\mathbf{y}_t$ is not guaranteed to be at $\mathfrak{X}$, hence the actual prediction of OLN is determined via a projection onto the set $\mathfrak{X}$, resulting with the vector $\mathbf{x}_{t+1} \in \mathfrak{X}$. However, similarly to ONS, the algorithm first transforms $\mathbf{y}_t$ via the (pseudo-)inverse of the matrix $A_t$ given by the sum of the outer products $\sum_{s=1}^t \nabla_s \nabla_s^\mathsf{T}$, and projections are taken with respect to $A_t$. In this context, we use the notation

$$\Pi_{\mathfrak{X}}^A(\mathbf{y}) = \arg\min_{\mathbf{x} \in \mathfrak{X}} (\mathbf{x} - \mathbf{y})^\mathsf{T} A (\mathbf{x} - \mathbf{y}).$$

to denote the projection onto a set $\mathfrak{X}$ with respect to the (semi-)norm $\|\cdot\|_A$ induced by a positive semidefinite matrix $A \succeq 0$.

---
**Algorithm 1** OLN: Online Lazy Newton
---
**Parameters:** initial point $\mathbf{x}_1 \in \mathcal{X}$, step size $\eta > 0$
Initialize $\mathbf{y}_0 = 0$ and $A_0 = 0$
**for** $t = 1, 2 \ldots T$ **do**
    Play $\mathbf{x}_t$, incur cost $f_t(\mathbf{x}_t)$, observe gradient $\nabla_t = \nabla f_t(\mathbf{x}_t)$
    Rank 1 update $A_t = A_{t-1} + \eta \nabla_t \nabla_t^\top$
    Online Newton step and projection:

$$\begin{aligned} \mathbf{y}_t &= \mathbf{y}_{t-1} - \nabla_t \\ \mathbf{x}_{t+1} &= \Pi_{\mathcal{X}}^{A_t}(A_t^\dagger \mathbf{y}_t) \end{aligned}$$

**end for**
**return**

---

The main motivation behind working with the $A_t$ as preconditioners is that—as demonstrated in our analysis—the algorithm becomes invariant to linear transformations of the gradient vectors $\nabla_t$. Indeed, if $P$ is some linear transformation, one can observe that if we run the algorithm on $P\nabla_t$ instead of $\nabla_t$, this will transform the solution at step $t$ from $\mathbf{x}_t$ to $P^{-1}\mathbf{x}_t$. In turn, the cumulative regret is invariant to such transformations. As seen in Theorem 1, this invariance indeed leads to an algorithm with an improved regret bound when the input representation of the data is poorly conditioned.

While the algorithm is very similar to ONS, it is different in several important aspects. First, unlike ONS, our *lazy* version maintains at each step a vector $\mathbf{y}_t$ which is updated without any projections. Projection is then applied only when we need to calculate $\mathbf{x}_{t+1}$. In that sense, it can be thought as a gradient descent method with lazy projections (analogous to dual-averaging methods) while ONS is similar to gradient descent methods with a greedy projection step (reminiscent of mirror-descent type algorithms). The effect of this, is a decoupling between past and future conditioning and projections: and if the transformation matrix $A_t$ changes between rounds, the lazy approach allows us to condition and project the problem at each iteration independently.

Second, ONS uses an initialization of $A_0 = \epsilon I_d$ (while OLN uses $A_0 = 0$) and, as a result, it is not invariant to affine transformations. While this difference might seem negligible as $\epsilon$ is typically chosen to be very small, recall that the matrices $A_t$ are used as preconditioners and their small eigenvalues can be very meaningful, especially when the problem at hand is poorly conditioned.

## 4   Application: Low Rank Experts

In this section we consider the Low-rank Experts problem and show how the Online Lazy Newton algorithm can be used to obtain a nearly optimal regret in that setting. In the Low-rank Experts problem, which is a variant of the classic problem of prediction with expert advice, a learner has to choose at each round $t = 1, \ldots, T$ between following the advice of one of $N$ experts. On round $t$, the learner chooses a distribution over the experts in the form of a probability vector $\mathbf{x}_t \in \Delta_n$ (here $\Delta_n$ denotes the $n$-dimensional simplex); thereafter, an adversary chooses a cost vector $\mathbf{g}_t \in [-1, 1]^N$ assigning losses to experts, and the player suffers a loss of $\mathbf{x}_t^\top \mathbf{g}_t \in [-1, 1]$. In contrast with the standard experts setting, here we assume that in hindsight the expert share a common low rank structure and we have that $\text{rank}(\mathbf{g}_1, \ldots, \mathbf{g}_T) \le r$, for some $r < N$.

It is known that in the stochastic setting (i.e., $\mathbf{g}_t$ are drawn i.i.d. from some fixed distribution) a follow-the-leader algorithm will enjoy a regret bound of $\min\{\sqrt{rT}, \sqrt{T \log N}\}$. In [15] the authors asked whether one can achieve the same regret bound in the online setting. Here we answer this question on the affirmative.

**Theorem 4** (Low Rank Experts). *Consider the low rank expert setting, where* $\text{rank}(\mathbf{g}_1, \ldots, \mathbf{g}_T) \le r$. *Set* $\eta = \sqrt{r \log(T)/T}$, *and run Algorithm 1 with* $\mathcal{X} = \Delta_n$ *and* $f_t(\mathbf{x}) = \mathbf{g}_t^\top \mathbf{x}$. *Then, the obtained regret satisfies*

$$\text{Regret}_T = O(\sqrt{rT \log T}).$$

This bound matches the $\Omega(\sqrt{rT})$ lower bound of [15] up to a $\log T$ factor, and improves upon their $O(r\sqrt{T})$ upper bound, so long as $T$ is not exponential in $r$. Put differently, if one aims at ensuring an

average regret of at most $\epsilon$, the OLN algorithm would need $O((r/\epsilon^2)\log(1/\epsilon))$ iterations as opposed to the $O(r^2/\epsilon^2)$ iterations required by the algorithm of [15]. We also remark that, since the Hedge algorithm can be used to obtain regret rate of $O(\sqrt{T \log N})$, we can obtain an algorithm with regret bound of the form $O\left(\min\{\sqrt{rT \log T}, \sqrt{T \log N}\}\right)$ by treating Hedge and OLN as meta-experts and applying Hedge over them.

## 5 Analysis

For the proofs of our main theorems we will rely on the following two technical lemmas.

**Lemma 5** ([17], Lemma 5). *Let $\Phi_1, \Phi_2 : \mathcal{X} \mapsto \mathbb{R}$ be two convex functions defined over a closed and convex domain $\mathcal{X} \subseteq \mathbb{R}^d$, and let $x_1 \in \arg\min_{x \in \mathcal{X}} \Phi_1(x)$ and $x_2 \in \arg\min_{x \in \mathcal{X}} \Phi_2(x)$. Assume that $\Phi_2$ is $\sigma$-strongly-convex with respect to a norm $\|\cdot\|$. Then, for $\phi = \Phi_2 - \Phi_1$ we have*

$$\|x_2 - x_1\| \le \frac{1}{\sigma}\|\nabla\phi(x_1)\|^*.$$

*Furthermore, if $\phi$ is convex then*

$$0 \le \phi(x_1) - \phi(x_2) \le \frac{1}{\sigma}\left(\|\nabla\phi(x_1)\|^*\right)^2.$$

The following lemma is a slight strengthening of a result given in [14].

**Lemma 6.** *Let $\mathbf{g}_1, \ldots, \mathbf{g}_T \in \mathbb{R}^d$ be a sequence of vectors, and define $G_t = H + \sum_{s=1}^t \mathbf{g}_s \mathbf{g}_s^\mathsf{T}$ for all $t$, where $H$ is a positive definite matrix such that $\|\mathbf{g}_t\|_H^* \le \gamma$ for all $t$. Then*

$$\sum_{t=1}^T \mathbf{g}_t^\mathsf{T} G_t^{-1} \mathbf{g}_t \le r \log\left(1 + \frac{\gamma^2 T}{r}\right),$$

*where $r$ is the rank of the matrix $\sum_{s=1}^t \mathbf{g}_s \mathbf{g}_s^\mathsf{T}$.*

*Proof.* Following [14], we first prove that

$$\sum_{t=1}^T \mathbf{g}_t^\mathsf{T} G_t^{-1} \mathbf{g}_t \le \log\frac{\det G_T}{\det H} = \log\det\left(H^{-1/2} G_T H^{-1/2}\right). \tag{5}$$

To this end, let $G_0 = H$, so that we have $G_t = G_{t-1} + \mathbf{g}_t \mathbf{g}_t^\mathsf{T}$ for all $t \ge 1$. The well-known matrix determinant lemma, which states that $\det(A - uu^\mathsf{T}) = (1 - u^\mathsf{T} A^{-1} u)\det(A)$, gives

$$\mathbf{g}_t^\mathsf{T} G_t^{-1} \mathbf{g}_t = 1 - \frac{\det(G_t - \mathbf{g}_t \mathbf{g}_t^\mathsf{T})}{\det G_t} = 1 - \frac{\det(G_{t-1})}{\det G_t}.$$

Using the inequality $1 - x \le \log(1/x)$ and summing over $t = 1, \ldots, T$, we obtain

$$\sum_{t=1}^T \mathbf{g}_t^\mathsf{T} G_t^{-1} \mathbf{g}_t \le \sum_{t=1}^T \log\frac{\det G_t}{\det G_{t-1}} = \log\frac{\det G_T}{\det H},$$

which yields Eq. (5).

Next, observe that $H^{-1/2} G_T H^{-1/2} = I + \sum_{s=1}^T H^{-1/2}\mathbf{g}_s \mathbf{g}_s^\mathsf{T} H^{-1/2}$ and

$$\mathrm{Tr}\left(\sum_{s=1}^T H^{-1/2}\mathbf{g}_s \mathbf{g}_s^\mathsf{T} H^{-1/2}\right) = \sum_{s=1}^T \mathrm{Tr}\left(\mathbf{g}_s^\mathsf{T} H^{-1}\mathbf{g}_s\right) = \sum_{s=1}^T (\|\mathbf{g}_s\|_H^*)^2 \le \gamma^2 T.$$

Also, the rank of the matrix $\sum_{s=1}^T H^{-1/2}\mathbf{g}_s \mathbf{g}_s^\mathsf{T} H^{-1/2} = H^{-1/2}(\sum_{s=1}^T \mathbf{g}_s \mathbf{g}_s^\mathsf{T})H^{-1/2}$ is at most $r$. Hence, all the eigenvalues of the matrix $H^{-1/2} G_T H^{-1/2}$ are equal to 1, except for $r$ of them whose sum is at most $r + \gamma^2 T$. Denote the latter by $\lambda_1, \ldots, \lambda_r$; using the concavity of $\log(\cdot)$ and Jensen's inequality, we conclude that

$$\log\det\left(H^{-1/2} G_T H^{-1/2}\right) = \sum_{i=1}^r \log\lambda_i \le r\log\left(\frac{1}{r}\sum_{i=1}^r \lambda_i\right) \le r\log\left(1 + \frac{\gamma^2 T}{r}\right),$$

which together with Eq. (5) gives the lemma. □

We can now prove our main results. We begin by proving Theorem 1.

***Proof of Theorem 1.*** For all $t$, let

$$\tilde{f}_t(\mathbf{x}) = \mathbf{g}_t^\top \mathbf{x} + \frac{\eta}{2}(\mathbf{g}_t^\top \mathbf{x})^2$$

and set

$$\widetilde{F}_t(\mathbf{x}) = \sum_{s=1}^{t} \tilde{f}_s(\mathbf{x}) = -\mathbf{y}_t^\top \mathbf{x} + \frac{1}{2}\mathbf{x}^\top A_t \mathbf{x}.$$

Observe that $\mathbf{x}_{t+1}$, which is the choice of Algorithm 1 at iteration $t+1$, is the minimizer of $\widetilde{F}_t$; indeed, since $y_t$ is in the column span of $A_t$, we can write up to a constant:

$$\widetilde{F}_t(\mathbf{x}) = \frac{1}{2}\left(\mathbf{x} - A_t^\dagger \mathbf{y}_t\right)^\top A_t \left(\mathbf{x} - A_t^\dagger \mathbf{y}_t\right) + \text{const.}$$

In other words, Algorithm 1 is equivalent to a follow-the-leader algorithm on the functions $\tilde{f}_t$.

Next, fix some positive definite matrix $H > 0$ and let $D_H = \max_{\mathbf{x},\mathbf{y}\in\mathcal{X}} \|\mathbf{x} - \mathbf{y}\|_H$ and $G_H = \max_{1\le t\le T} \|\mathbf{g}_t\|_H^*$. Next we have

$$\widetilde{F}_t(\mathbf{x}) + \tfrac{\eta}{2}\|\mathbf{x} - \mathbf{x}_{t+1}\|_H^2 =$$

$$= \frac{1}{2}\mathbf{x}^\top A_t \mathbf{x} - \mathbf{y}_t^\top \mathbf{x} + \tfrac{\eta}{2}\|\mathbf{x} - \mathbf{x}_{t+1}\|_H^2$$

$$= \frac{1}{2}\|\mathbf{x}\|_{A_t}^2 + \frac{\eta}{2}\|\mathbf{x}\|_H^2 - \mathbf{y}_t^\top \mathbf{x} - 2\mathbf{x}_{t+1}^\top \mathbf{x} + \|\mathbf{x}_{t+1}\|_H^2$$

$$= \frac{\eta}{2}\|\mathbf{x}\|_{G_t}^2 - \mathbf{y}_t^\top \mathbf{x} - 2\mathbf{x}_{t+1}^\top \mathbf{x} + \|\mathbf{x}_{t+1}\|_H^2,$$

where $G_t = \sum_{s=1}^{t} \mathbf{g}_t \mathbf{g}_t^\top + H$.

In turn, we have that the function is $\eta$-strongly convex with respect to the norm $\|\cdot\|_{G_t}$, where $G_t = H + \sum \mathbf{g}_t \mathbf{g}_t^\top$, and is minimized at $\mathbf{x} = \mathbf{x}_{t+1}$. Then by Lemma 5 with $\Phi_1(\mathbf{x}) = \widetilde{F}_{t-1}(\mathbf{x})$ and $\Phi_2(\mathbf{x}) = \widetilde{F}_t(\mathbf{x}) + \tfrac{\eta}{2}\|\mathbf{x} - \mathbf{x}_{t+1}\|_H^2$, thus $\phi(\mathbf{x}) = \tilde{f}_t(\mathbf{x}) + \tfrac{\eta}{2}\|\mathbf{x} - \mathbf{x}_{t+1}\|_H^2$, we have

$$\tilde{f}_t(\mathbf{x}_t) - \tilde{f}_t(\mathbf{x}_{t+1}) + \frac{\eta}{2}\|\mathbf{x}_t - \mathbf{x}_{t+1}\|_H^2$$

$$\le \frac{1}{\eta}(\|\mathbf{g}_t + \eta\mathbf{g}_t\mathbf{g}_t^\top \mathbf{x}_t + \eta H(\mathbf{x}_t - \mathbf{x}_{t+1})\|_{G_t}^*)^2$$

$$\le \frac{2}{\eta}(1 + \eta\mathbf{g}_t^\top \mathbf{x}_t)^2(\|\mathbf{g}_t\|_{G_t}^*)^2 + 2\eta(\|H(\mathbf{x}_t - \mathbf{x}_{t+1})\|_{G_t}^*)^2 \qquad \because \ \|\mathbf{v} + \mathbf{u}\|^2 \le 2\|\mathbf{v}\|^2 + 2\|\mathbf{u}\|^2$$

$$\le \frac{8}{\eta}(\|\mathbf{g}_t\|_{G_t}^*)^2 + 2\eta(\|H(\mathbf{x}_t - \mathbf{x}_{t+1})\|_{G_t}^*)^2 \qquad \because \ \frac{1}{\eta} \ge \max_{\mathbf{x}\in\mathcal{X}} |\mathbf{g}_t^\top \mathbf{x}|$$

$$\le \frac{8}{\eta}(\|\mathbf{g}_t\|_{G_t}^*)^2 + 2\eta(\|H(\mathbf{x}_t - \mathbf{x}_{t+1})\|_H^*)^2 \qquad \because \ H \prec G_t \Rightarrow H^{-1} \succ G_t^{-1}$$

$$= \frac{8}{\eta}(\|\mathbf{g}_t\|_{G_t}^*)^2 + 2\eta\|\mathbf{x}_t - \mathbf{x}_{t+1}\|_H^2.$$

Overall, we obtain

$$\sum_{t=1}^{T} \tilde{f}_t(\mathbf{x}_t) - \tilde{f}_t(\mathbf{x}_{t+1}) \le \frac{8}{\eta}\sum_{t=1}^{T} \mathbf{g}_t^\top G_t^{-1}\mathbf{g}_t + \frac{3\eta}{2}\sum_{t=1}^{T} \|\mathbf{x}_t - \mathbf{x}_{t+1}\|_H^2.$$

By the FTL-BTL Lemma (e.g., [16]), we have that $\sum_{t=1}^{T} \tilde{f}_t(\mathbf{x}_t) - \tilde{f}_t(\mathbf{x}^\star) \le \sum_{t=1}^{T} \tilde{f}_t(\mathbf{x}_t) - \tilde{f}_t(\mathbf{x}_{t+1})$. Hence, we obtain that:

$$\sum_{t=1}^{T} \tilde{f}_t(\mathbf{x}_t) - \tilde{f}_t(\mathbf{x}^\star) \le \frac{8}{\eta}\sum_{t=1}^{T} \mathbf{g}_t^\top G_t^{-1}\mathbf{g}_t + \frac{3\eta}{2}\sum_{t=1}^{T} \|\mathbf{x}_t - \mathbf{x}_{t+1}\|_H^2.$$

Plugging in $f_t(\mathbf{x}) = \mathbf{g}_t^\top \mathbf{x} + \frac{\eta}{2}(\mathbf{g}_t^\top \mathbf{x})^2$ and rearranging, we obtain

$$\sum_{t=1}^{T} \mathbf{g}_t^\top (\mathbf{x}_t - \mathbf{x}^\star) \leq \frac{8}{\eta} \sum_{t=1}^{T} \mathbf{g}_t^\top G_t^{-1} \mathbf{g}_t + \frac{3\eta}{2} \sum_{t=1}^{T} \|\mathbf{x}_t - \mathbf{x}_{t+1}\|_H^2 + \frac{\eta}{2} \sum_{t=1}^{T} (\mathbf{g}_t^\top \mathbf{x}^\star)^2$$

$$\leq \frac{8}{\eta} \sum_{t=1}^{T} \mathbf{g}_t^\top G_t^{-1} \mathbf{g}_t + \frac{\eta}{2} \sum_{t=1}^{T} (\mathbf{g}_t^\top \mathbf{x}^\star)^2 + \frac{3\eta}{2} T D_H^2$$

$$\leq \frac{8r}{\eta} \log\Big(1 + \frac{G_H^2 T}{r}\Big) + \frac{3\eta}{2} T D_H^2 + \frac{\eta}{2} \sum_{t=1}^{T} (\mathbf{g}_t^\top \mathbf{x}^\star)^2,$$

Finally, note that we have obtained the last inequality for every matrix $H > 0$. By rescaling a matrix $H$ and re-parametrizing $H \to H/(\sqrt{T}D_H)$ we obtain a matrix whose diameter is $D_H \to 1/\sqrt{T}$ and $G_H \to \sqrt{T}D_H G_H$. Plugging these into the last inequality yield the result. $\qquad\square$

***Proof of Theorem 3.*** To simplify notations, let us assume that $|\nabla_t^\top \mathbf{x}^*| \leq 1$. We get from Corollary 2 that for every $\eta$:

$$\mathrm{Regret}_T \leq \frac{2r}{\eta} \log\Big(1 + \frac{D^2 G^2 T^2}{r}\Big) + 3\eta(1 + T).$$

For each $G_H$ and $D_H$ we can set $\eta = \sqrt{(2r/T) \log(1 + G_H^2 D_H^2/r)}$ and obtain the regret bound

$$\mathrm{Regret}_T \leq \sqrt{rT \log\Big(1 + \frac{D_H^2 G_H^2 T}{r}\Big)}.$$

Hence, we only need to show that there exists a matrix $H > 0$ such that $D_H^2 G_H^2 = O(r)$. Indeed, set $S = \mathrm{span}(\nabla_1, \ldots, \nabla_T)$, and denote $\mathfrak{X}_S$ to be the projection of $\mathfrak{X}$ onto $S$ (i.e., $\mathfrak{X}_S = P\mathfrak{X}$ where $P$ is the projection over $S$). Define

$$\mathcal{B} = \{\nabla \in S : |\nabla^\top \mathbf{x}| \leq 1, \ \ \forall \mathbf{x} \in \mathfrak{X}_S\}.$$

Note that by definition we have that $\{\nabla_t\}_{t=1}^{T} \subseteq \mathcal{B}$. Further, $\mathcal{B}$ is a symmetric convex set, hence by an ellipsoid approximation we obtain a positive semidefinite matrix $B \geq 0$, with positive eigenvalues restricted to $S$, such that

$$\mathcal{B} \subseteq \{\nabla \in S : \nabla^\top B \nabla \leq 1\} \subseteq r\mathcal{B}.$$

By duality we have that

$$\tfrac{1}{r} \mathfrak{X}_S \subseteq \tfrac{1}{r} \mathcal{B}^* \subseteq \{\mathbf{v} \in S : \mathbf{v}^\top B^\dagger \mathbf{v} \leq 1\}.$$

Thus if $P_S$ is the projection over $S$ we have for every $\mathbf{x} \in \mathfrak{X}$ that $\mathbf{x}^\top P_S B^\dagger P_S \mathbf{x} \leq r$. On the other hand for every $\nabla_t$ we have $\nabla_t^\top B \nabla_t \leq 1$. We can now choose $H = B^\dagger + \epsilon I_d$ where $\epsilon$ is arbitrary small and have

$$\nabla_t^\top H^{-1} \nabla_t = \nabla_t^\top (B^\dagger + \epsilon I_d)^{-1} \nabla_t \leq 2$$

and

$$\mathbf{x}^\top H \mathbf{x} = \mathbf{x}^\top P_S^\top B^\dagger P_S \mathbf{x} + \epsilon \|\mathbf{x}\|^2 \leq 2r. \qquad\square$$

## Acknowledgements

The authors would like to thank Elad Hazan for helpful discussions. RL is supported by the Eric and Wendy Schmidt Fund for Strategic Innovations.

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
