[Reviews · NeurIPS 2017]

Reviewer 1



Summary: This paper proposes a new optimization algorithm, Online Lazy Newton (OLN), based on Online Newton Step (ONS) algorithm. Unlike ONS which tries to utilize curvature information within convex functions, OLN aims at optimizing general convex functions with no curvature. Additionally, by making use of low rank structure of the conditioning matrix, the authors showed that OLN yields better regret bound under certain conditions. Overall, the problem is well-motivated and the paper is easy to follow. Major Comments: 1. The major difference between OLN and ONS is that ONS introduces a lazy evaluation step, which accumulates the negative gradients at each round. The authors claimed in Lines 155-157 that this helps in decoupling between past and future conditioning and projections and it is better in the case when transformation matrix is changing between rounds. It would be better to provide some explanations. 2. Lines 158-161, it is claimed that ONS is not invariant to affine transformation due to its initialization. In my understanding, the regularization term is added partly because it allows for an invertible matrix and can be omitted if Moore-Penrose pseudo-inverse is used as in the FTAL. 3. Line 80 and Line 180, it is claimed that O(\sqrt(r T logT)) is improved upon O(r \sqrt(T)). The statement will hold under the condition that r/logT = O(1), is this always true? Minor: 1. Lines 77-79, actually both ONS and OLN utilizes first-order information to approximate second-order statistics. 2. Line 95, there is no definition of the derivative on the right-hand-side of the equation prior to the equation. 3. Line 148-149, on the improved regret bound. Assuming that there is a low rank structure for ONS (similar to what is assumed in OLS), would the regret bound for OLS still be better than ONS? 4. Line 166, 'In The...' -> 'In the...' 5. Line 181, better to add a reference for Hedge algorithm. 6. Line 199, what is 'h' in '... if h is convex ...'? 7. Line 211-212, '...all eigenvalues are equal to 1, except for r of them...', why it is the case? For D = I + B defined in Line 209, the rank for B is at most r, then at least r of the eigenvalues of D are equal to 1. The regret bound depends on the low rank structure of matrix A as in Theorem 3, another direction that would be interesting to explore is to consider the low rank approximation to the matrix A and check if similar regret bound can be derived, or under which conditions similar regret bound can be derived. I believe the proposed methods will be applicable to more general cases along this direction.

Reviewer 2



This paper considers online learning problem with linear and low-rank loss space, which is an interesting and relative new topic. The main contribution lies in proposing an online newton method with a better regret than [Hazan et. al. 2016], namely O(\sqrt(rT logT) vs. O(r\sqrt(T)). And the analysis is simple and easy to follow. There are a few concerns listed below. 1. The better regret bound is arguable. Assuming low rank, ‘r’ is typical small while ‘T’ could be very large in online setting. Thus, comparing with existing work, which establishes regrets of O(r\sqrt(T)) and O(\sqrt(rT)+logNlogr)), the achieved result is not very exciting. 2. Algorithm 1 requires a matrix inverse and solving a non-linear programming per iteration, and a sub-routine appears inevitable for most problems (e.g. the low rank expert example with domain `simplex’). Such complexity prevents the algorithm from real online applications and limits it in analysis. 3. The reviewer appreciates strong theoretical work without experiments. However, the presented analysis of this paper is not convincing enough under NIPS criterion. A empirical comparison with AdaGrad and [Hazan et. al. 2016] would be a nice plus. 5. `A_t^{-1}’ in Algorithm 1 is not invertible in general. Is it a Moore–Penrose pseudoinverse? And does a pseudoinverse lead to a failure in analysis? 6. Though the authors claim Algorithm 1 is similar to ONS, I feel it is closer to `follow the approximate leader (ver2)’ in [Hazan et.al. 2006]. Further discussion is desirable. Overall, this work makes a theoretical step in special online setting and may be interesting to audiences working in this narrow direction. But it is not very exciting to general optimization community and appears too expensive in practice. ************* I read the rebuttal and removed the comment on comparison with recent work on arxiv. Nevertheless, I'm still feel that O(\sqrt(rT logT) is not a strong improvement over O(r\sqrt(T)), given that T is much larger than r. An ideal answer to the open problem should be O(\sqrt(rT) or a lower bound showing that O(\sqrt(rT logT) is inevitable.

Reviewer 3



The paper analyzes a particular variant of online Newton algorithm for online linear(!) optimization. Using second-order algorithm might sound non-sensical. However, the point of the paper is to compete with the best preconditioning of the data. The main reason for doing this is to solve the low rank expert problem. (In the low-rank expert problem, the whole point is to find a basis of the loss matrix.) The main result of the paper is an algorithm for low rank expert problem that has regret within sqrt(log T) of the lower bound for the low rank experts problem. In particular, it improves sqrt{r} factor on the previous algorithm. Large parts of the analysis are the same as in Vovk-Azoury-Warmuth forecaster for online least squares (see e.g. the book Gabor Lugosi & Nicolo Cesa-Bianchi), the second-order Perceptron ("A second-order perceptron " by Nicolo Cesa-Bianchi, Alex Conconi, and Claudio Gentile), or the analysis of Hazan & Kale for exp-concave functions. I suggest that authors reference these papers. The second-order perceptron paper is particularly relevant since it's a classification problem, so there is no obvious second-order information to use, same as in the present paper. Also, the paper "Efficient Second Order Online Learning by Sketching" by Haipeng Luo, Alekh Agarwal, Nicolo Cesa-Bianchi, John Langford is worth mentioning, since it also analyzes online Newton's method that is affine invariant i.e. the estimate of the Hessian starts with zero (see https://arxiv.org/pdf/1602.02202.pdf Appendix D). The paper is nicely written. I've spot-checked the proofs. They look correct. At some places (Page 7, proof of Theorem 1), inverse of non-invertible matrix A_t is used. It's not a big mistake, since a pseudo-inverse can be used instead. However, this issue needs to be fixed before publication.